# MathGAP: Out-of-Distribution Evaluation on Problems with Arbitrarily Complex Proofs

**Andreas Opedal**[1,2,*]   **Haruki Shirakami**[1,3,*]
**Bernhard Schölkopf**[1,2]    **Abulhair Saparov**[4]    **Mrinmaya Sachan**[1]
[1]ETH Zürich    [2]Max Planck Institute for Intelligent Systems, Tübingen
[3]Idiap Research Institute    [4]Purdue University
andreas.opedal@inf.ethz.ch   haruki.shirakami@idiap.ch

## Abstract

Large language models (LLMs) can solve arithmetic word problems with high accuracy, but little is known about how well they generalize to more complex problems. This is difficult to study, as (i) much of the available evaluation data has already been seen by the most capable models during training, and (ii) existing benchmarks do not capture how problem proofs may be arbitrarily complex in various ways. In this paper, we present a data-generation framework for evaluating LLMs on problems with arbitrarily complex arithmetic proofs, called MathGAP. MathGAP generates problem statements and chain-of-thought reasoning traces according to specifications about their arithmetic proof structure, enabling systematic studies on easy-to-hard generalization with respect to complexity of proof trees. Using Math-GAP, we find that LLMs show a significant decrease in performance as proofs get deeper and wider. This effect is more pronounced in complex, nonlinear proof structures, which are challenging even for the most capable models. The models are also sensitive to simple changes in sentence ordering. However, they remain capable of solving *some* complex problems, suggesting that reasoning generalization is noisy.

 https://github.com/eth-lre/mathgap-experiments

## 1 Introduction

High performance on reasoning benchmarks is often taken as evidence that transformer-based large language models (LLMs) can "reason". However, many current benchmarks are unreliable, as it is likely that the problems they contain are present in the model training data (Sainz et al., 2023; Deng et al., 2024; Zhang et al., 2024). Moreover, most evaluations fail to capture that reasoning problems can be arbitrarily complex through composition of subproofs and use of multiple rules of inference. High accuracy on a specific set of problems in a benchmark dataset is therefore not sufficient to conclude that LLMs can generalize to more complex, unseen problems. To obtain a more appropriate empirical measure of reasoning ability, one must evaluate on data that is *not* present in any benchmark dataset, containing proofs that are *more* complex than the ones the model has already seen.

In this paper, we aim to satisfy these desiderata. We consider math word problems (MWPs), which are one of the most frequently used testbeds for LLM reasoning, and propose a framework for evaluating Mathematical Generalization on Arithmetic Proofs—**MathGAP**. We start by discussing how solutions to MWPs can be characterized in terms of proof trees, in which nodes are labeled with logical statements about the problem under a world-model framework (Opedal et al., 2023), and the edges are induced by proof steps on such logical statements. Doing so reveals several ways of characterizing a problem's complexity based on its proof tree, e.g., depth, width, shape, and the ordering of nodes (all defined in §3.2). With this formalism, we develop a method to generate problems with specified proof tree characteristics. Because this method is based on proof trees, it also enables rich annotations in the form of ground-truth reasoning traces. More specifically, traversing the proof trees in post order, we obtain chain-of-thought (CoT; Wei et al., 2022) reasoning traces in an automated manner. Fig. 1 illustrates the generation method, showing a proof tree having depth 2, width 5, what we call *nonlinear* shape, and where the ordering of the sentences is given by traversing the leaf nodes in left-to-right order.

---

*Equal contribution.

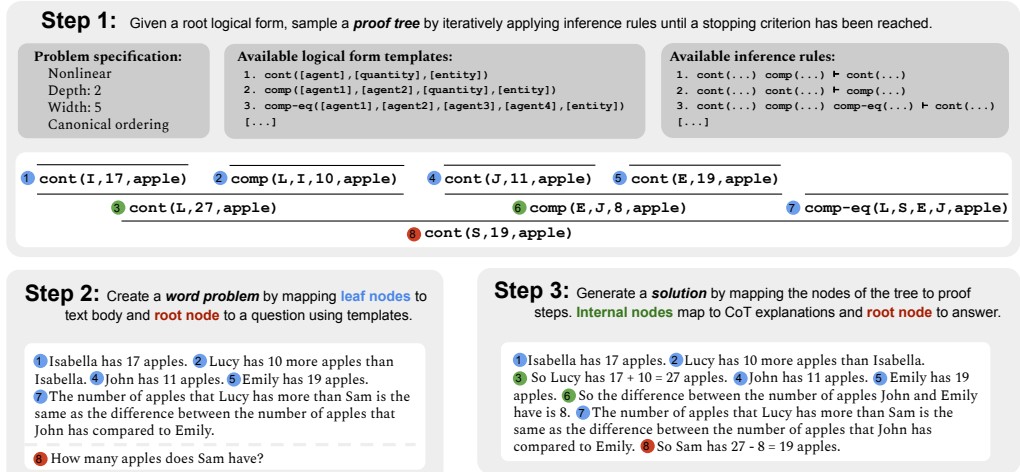

Figure 1: We propose MathGAP, an evaluation framework for arithmetic reasoning in which LLMs are tested on problems with proofs of arbitrary complexity. This diagram shows how problems and CoT solution annotations are generated under our formalism. The complete list of logical forms and inference rules that we consider in our experiments are given in Tables 1 and 2, respectively.

This generation method forms the backbone of MathGAP, our framework for studying out-of-distribution (OOD) generalization gaps on arithmetic proofs. MathGAP avoids data contamination since the distribution of generated test problems can always be different from those in training or in context, by design. When the performance on problems at one level of complexity hits saturation, we can flexibly generate a new set of problems that are even more complex. This is in spirit similar to Dynabench (Kiela et al., 2021) but, importantly, does not require human annotators.

MathGAP enables many new empirical investigations. In this paper, we perform a systematic analysis on whether LLMs can use simple examples in context to solve more complex ones at inference. To this end, we consider two different distributions of in-context CoT demonstrations: examples with only the simplest possible proof trees, and a range of examples with varying complexity, akin to curriculum learning (Bengio et al., 2009). We then evaluate on test sets containing problems that are increasingly more complex than the in-context examples. The performance is further compared to in-distribution and zero-shot baselines. Our study includes Mixtral-8x7B (Jiang et al., 2024a), Llama3-8B, Llama3-70B (Llama Team, 2024), GPT-3.5-Turbo, GPT-4o (OpenAI, 2024), and a few additional evaluations on o1-preview and DeepSeek-R1 (DeepSeek-AI, 2025). We present several findings:

i) Generalization to proof width is harder than to proof depth, but there is a steady decrease in performance as proofs get both deeper and wider. The performance decrease is particularly notable for the so-called nonlinear problems; solving them requires keeping intermediate steps longer in memory than for linear problems, for which intermediate steps can be consumed immediately (under post-order traversal). Even GPT-4o fails to solve the most complex nonlinear problems.

ii) However, the performance decrease is moderate in most cases. This suggests that the models are able to generalize to some extent, albeit with a noisy form of reasoning (Prabhakar et al., 2024).

iii) All models are sensitive to a simple change in sentence ordering, in which one sentence is moved to the front of the problem. Moreover, we find that the problem is easiest for LLMs if the sentence is moved from the beginning or the end rather than from the middle of the problem. This is surprising since one would expect a monotonically decreasing relationship between accuracy and the distance of movement.

iv) We do not see a consistent advantage to using in-context examples. Contrary to our expectations, in-distribution examples are not always beneficial for performance, as compared to zero-shot. This suggests that some problems are hard to learn even when the LLM is given demonstrations of similar problems, independently of its generalization ability. Demonstrating a range of examples with varying complexity is usually preferable to demonstrating simple examples, suggesting that LLMs benefit from a diverse prompt.

v) o1 and R1 outperform the others on nonlinear problems. However, we are able to create a test set on which they solve only 5% and 11% respectively—indicating that MathGAP is future-proof.

## 2 RELATED WORK

**Data contamination.** Machine learning, broadly, is the study of algorithms that can learn from data and generalize to *unseen* data. Machine learning models must therefore be evaluated on unseen test sets. As it turns out, modern-day LLMs are not being properly evaluated in this regard. Much of the data they are evaluated on has already been used during model training, both labeled (Dodge et al., 2021; Deng et al., 2024) and unlabeled (Elazar et al., 2024). Researchers have raised serious concerns about such data contamination issues (Jacovi et al., 2023; Sainz et al., 2023). For arithmetic reasoning specifically, Zhang et al. (2024) recently found evidence that the widely used GSM8k benchmark (Cobbe et al., 2021) is contaminated and presented evaluations on a new dataset that is not publicly released. MathGAP mitigates data contamination by creating new, more complex synthetic test sets.

**Generalization on reasoning tasks.** Generalization to data from a different distribution (i.e., OOD data) is a core problem in machine learning and has also been studied extensively in the context of LLM reasoning (Schwarzschild et al., 2021; An et al., 2023; Kudo et al., 2023; Liu et al., 2023; Saparov & He, 2023; Zhang et al., 2023; Mészáros et al., 2024; Thomm et al., 2024). While LLMs' performance on OOD data can be improved by various techniques (Anil et al., 2022; Borazjanizadeh & Piantadosi, 2024; Hu et al., 2024), generalization to problems of arbitrary complexity under a set of known inference rules is still an open problem. For instance, transformer-based LLMs struggle to generalize to longer problem proofs when solving logical reasoning tasks (Dziri et al., 2023; Saparov et al., 2023). We contribute to this literature by providing an evaluation framework for OOD generalization to complex arithmetic proofs. Our work relates to scalable oversight (Bowman et al., 2022) and easy-to-hard generalization (Burns et al., 2023; Hase et al., 2024; Sun et al., 2024), which can be studied in a principled manner with MathGAP by characterizing what is easy and what is hard.

**Evaluation on MWP benchmarks.** MWPs are often treated as a testbed for studying reasoning abilities of LLMs (Patel et al., 2021; Fu et al., 2023; Shakarian et al., 2023; Stolfo et al., 2023; Zong & Krishnamachari, 2023; Huang et al., 2024; Yu et al., 2024; Ye et al., 2025); such problems are interesting because they require several distinct skills to solve, while remaining conceptually simple (Stern, 1993). However, we are not aware of much work on generalization in regard to problem complexity in this domain. Our study in §5 is related to Hase et al. (2024), who fine-tune LLMs on easy problems and evaluate them on harder ones at inference, using GSM8k and other possibly contaminated datasets. Their complexity metrics focus on problem length, but do not capture *how* a reasoning problem gets more complex (i.e., through the characteristics of the proof tree). Mirzadeh et al. (2025) recently performed evaluations on a version of GSM8k in which variable names and numbers have been altered, and found that LLMs are surprisingly sensitive to such substitutions. MathGAP is more general; while it allows for such substitutions as well, our main goal is to vary the proof structures.

## 3 A FORMAL TREATMENT OF MATH WORD PROBLEMS

We aim to generate new problems of arbitrary complexity, so we require a formalism to characterize complexity in a precise manner. First, we explain how the semantics expressed in MWPs can be written as sequences of logical forms (§3.1). We then show how to apply inference rules on these logical forms to deduce new ones (§3.2). This leads to a view of problem solutions as proof trees, the structure of which can be used to characterize the reasoning required to solve the problem.

### 3.1 MATH WORD PROBLEMS AS LOGICAL FORMS

Each MWP is represented as a sequence of logical forms under the formalism from Opedal et al. (2023; 2024), which is a subset of first-order logic. A logical form is a truth statement about the world that is being described by the problem, representing some arithmetic relationship between the number of entities possessed by one or several agents. Such statements can refer to the number of entities an agent has, how many more entities an agent has relative to another, the action of an agent giving their entities to another agent, etc. Formally, a logical form represents the semantics of a single sentence and consists of a **predicate** that takes a set of **properties** as arguments. Every logical form has an **agent** property, an **entity** property, and a **quantity** property. Other, optional properties include **unit** and **attribute**. We define a **world model** as a sequence of logical forms that align with the sentences of a problem text, thus representing the semantics of the whole problem.

Table 1: Example logical forms. A logical form comprises a predicate and a set of properties specific to that predicate. Each logical form induces natural language sentences that preserve its semantics.

| Logical Form | | Example Templates | Example Sentences |
|---|---|---|---|
| **Predicate (abbr.)** | **Properties** | | |
| `container` `(cont)` | `(agent=a,` `quantity=q,` `entity=e,` `attribute=k,` `unit=u)` | *[a] has [q] [u]s of [k] [e]s.* | *Alice has 5 kilograms of red apples.* |
| | | *[a] owns [q] [u]s of [k] [e]s.* | *Alice owns 5 kilograms of red apples.* |
| `comparison` `(comp)` | `(agentA=a,` `agentB=b,` `quantity=q,` `entity=e)` | *[b] has [q] fewer [e]s than [a].* | *Bob has 3 fewer apples than Alice.* |
| | | *[a] has [q] more [e]s than [b].* | *Alice has 3 more apples than Bob.* |
| `transfer` | `(receiver_agent=b,` `sender_agent=a,` `quantity=q,` `entity=e)` | *[a] gave [b] [q] [e]s.* | *Alice gave Bob 3 apples.* |
| | | *[b] got [q] more [e]s from [a].* | *Bob got 3 more apples from Alice.* |
| `partwhole` | `(whole_agent=`$\wedge_{i=1}^{n} a_i,$ `quantity=q,` `whole_entity=e)` | *[a₁], ..., and [aₙ] have [q] [e]s combined.* | *Alice and Bob have 8 fruits combined.* |
| `comp-eq` | `(agentA=a,` `agentB=b,` `agentC=c,` `agentD=d,` `entity=e)` | *The number of [e]s that [c] has more than [d] is equal to the difference between the number of [e]s that [a] and [b] have.* | *The number of apples that Charlie has more than David is equal to the difference between the number of apples that Alice and Bob have.* |

For a given predicate and set of properties, we write `predicate(property1=x,` `property2=y, ...)` to express a logical form. Property names are, however, often omitted for brevity, as in `predicate(x, y, ...)`. There are two types of predicates: (i) those that represent the ownership of some entity by an agent, called `container` (abbreviated `cont`), and (ii) those that represent arithmetic relationships between two sets of properties. The relationship predicates correspond to arithmetic concepts and are based on a taxonomy from the learning sciences (Riley et al., 1983); we use `transfer`, `comparison` (abbreviated `comp`), `partwhole` and `comp-eq`.[1] The `comp-eq` predicate was introduced in this paper in order to construct nonlinear problems of arbitrary length—see App. D for details. Table 1 gives concrete example sentences for logical forms with each of these predicates, Fig. 5 shows the world model representation of the problem in Fig. 1, and App. A provides further intuition.[2]

The logical forms in the world model of a problem are separated into **body** and **question**. The question is a single logical form that appears at the end of the world model. It holds a placeholder variable instead of an explicit quantity, which represents the correct answer to the problem. For example, the interrogative sentence *"How many apples does Alice have?"* is represented by the question `cont(Alice, q, apples)`, where `q` is a placeholder variable. The body consists of all logical forms in the world model that are not questions. The **answer** to a problem is the logical form declared by the question with the correct numerical quantity substituted into its placeholder variable.

### 3.2 CHAIN-OF-THOUGHT SOLUTIONS AS PROOF TREES

Solving MWPs is a form of deductive reasoning, where the correct answer follows from what is stated in the text through a combination of rules of arithmetic and world knowledge. We use such derivations, called proofs, to characterize these problems. The proofs in our setup are analogous to those in other proof systems, such as natural deduction (Gentzen, 1935; Pfenning, 2004).[3]

We want to formally reason over the logical forms introduced in §3.1. To that end we introduce a set of **inference rules**, which are used to prove new logical forms from previously known ones. A **proof step**, written in Gentzen-style notation as

$$\frac{L_1 \quad L_2 \quad \cdots \quad L_N}{L},$$

---

[1] The transfer and part-whole concepts are usually called *change* and *combine*, respectively, in learning science literature (Nesher et al., 1982).

[2] In order to denote that the items of several agents are combined under a `partwhole`, we treat the conjunction of agents $\wedge_{i=1}^{n} a_i = a$ as another agent in Table 1. The same is done for entities, as shown in Table 2.

[3] In fact, each of our inference rules shown in Table 2 can be decomposed into the primitive inference rules of natural deduction, using first-order logic expressions. App. B demonstrates by example how this is done.

Table 2: Inference rule templates (**left**) with corresponding examples of rule applications in natural language (**right**). The variables `a`, `b`, `c`, `d` refer to agents, `q`, $q_1$, $q_2$ refer to quantities, and `e`, `f` refer to entities. Attribute and unit properties are excluded here, but analogous rules exist in which they are present. All inference rules are commutative except the `transfer` rule; see footnote 4. Rule applications of an inference rule with premises $L_1, L_2, \ldots, L_N$ and consequent $L$ are written as $L'_1. L'_2. \ldots . L'_N. \vdash L'.$, where $L'$ is a natural language expression for the logical form $L$. Axioms are formed from the facts stated in the problem, as illustrated in Fig. 1.

| Inference Rules | Example Sentences |
|---|---|
| $$\frac{\text{cont(a, }q_1\text{, e)} \quad \text{comp(b, a, }q_2\text{, e)}}{\text{cont(b, }q_1 + q_2\text{, e)}}$$ | *Alice has 3 apples. Bob has 2 more apples than Alice.* $\vdash$ *Bob has 5 apples.* |
| $$\frac{\text{cont(a, }q_1\text{, e)} \quad \text{transfer(a, b, }q_2\text{, e)}}{\text{cont(a, }q_1 + q_2\text{, e)}}$$ | *Alice has 3 apples. Bob gave 2 apples to Alice.* $\vdash$ *Alice has 5 apples.* |
| $$\frac{\text{cont(a, }q_1\text{, e)} \quad \text{cont(b, }q_2\text{, e)}}{\text{comp(b, a, }q_2 - q_1\text{, e)}}$$ | *Alice has 3 apples. Bob has 5 apples.* $\vdash$ *Bob has 2 more apples than Alice.* |
| $$\frac{\text{cont(}a_1\text{, }q_1\text{, }e_1\text{)} \quad \ldots \quad \text{cont(}a_n\text{, }q_n\text{, }e_n\text{)}}{\text{partwhole(}\wedge_{i=1}^{n}a_i\text{, }\sum_{i=1}^{n}q_i\text{, }\wedge_{i=1}^{n}e_i\text{)}}$$ | *Alice has 3 apples. Bob has 5 apples.* $\vdash$ *Alice and Bob have 8 fruits combined.* |
| $$\frac{\text{cont(a, }q_1\text{, e)} \quad \text{comp(d, c, }q_2\text{, e)} \quad \text{comp-eq(b, a, d, c)}}{\text{cont(b, }q_1 + q_2\text{, e)}}$$ | *Alice has 7 apples. David has 2 more apples than Charlie. The number of apples that Bob has more than Alice is the same as the difference between the number of apples that David and Charlie have.* $\vdash$ *Bob has 9 apples.* |

is an instance of an inference rule where the **conclusion** logical form $L$ is deduced from **premises** $L_1, L_2, \ldots, L_N$. An **axiom** is a proof step without premises. The axioms for a problem are the logical forms in the body of its world model, i.e., all logical forms excluding the question. Table 2 lists the inference rules that we use, along with examples of how they may be expressed in natural language.[4] For example, consider the two logical forms `cont(Isabella, 17, apple)` and `comp(Lucy, Isabella, 10, apple)` from Fig. 1, representing the facts that Isabella has 17 apples and Lucy has 10 more apples than Isabella, respectively. The proof step

$$\frac{\text{cont(Isabella, 17, apple)} \quad \text{comp(Lucy, Isabella, 10, apple)}}{\text{cont(Lucy, 17 + 10, apple)}}$$

lets us deduce the logical form `cont(Lucy, 27, apple)`, i.e., that Lucy has 27 apples.

A math word problem can be solved by applying inference rules until the answer to the problem has been proved. Formally, a **proof tree** is a rooted ordered tree where each node is labeled with a unique logical form $L$, which is the conclusion of a proof step with premises $L_1, \ldots, L_N$, matching the labels of the child nodes in the same order. Leaf nodes are thus labeled with axioms. A **proof** of a *particular problem* is a proof tree where the labels on the leaf nodes are the logical forms in the body of the problem, and the label at the root is the answer to the problem. The tree shown in Fig. 1 illustrates a proof of the problem given in the same figure.

**Complexity of proof trees.** We seek a systematic way to analyze whether LLMs can generalize from simple to more complex proofs.[5] The definition of proof given above gives rise to several ways of characterizing reasoning, out of which we consider four: depth, width, shape, and ordering. The **depth** of a proof tree is defined as its height, i.e., the maximum path length from the root node to any leaf node. Next, the **width** of a proof tree is the number of leaf nodes, or axioms, that it contains. The proof tree in Fig. 1 thus has depth 2 and width 5. In words, this means that the answer is two proof steps "away"

---

[4]The list is not exhaustive as additional variations on the rules exist. In particular, the logical forms may contain attributes and units (such as the `cont` examples in Table 1), the `cont` premises may be replaced by the `partwhole` premises (so that `partwhole` can be used for further proof steps), and the premises can be given in any order for all inference rules (i.e., they are commutative) except the one using `transfer`. The `transfer` rule is non-commutative since it involves a notion of time. For instance, "Alice (now) has 5 apples. Alice ate 2 apples." does not imply the same conclusion as "Alice ate 2 apples. Alice (now) has 5 apples." Moreover, note that the third inference rule listed in Table 2 is commutative, since swapping the two premises yields the conclusion `comp(a, b, `$q_1 - q_2$`, e)`, which is equivalent to `comp(b, a, `$q_2 - q_1$`, e)`.

[5]Note that our use of the term "complexity" is different from its conventional use in proof theory, in which proof complexity refers to the efficiency of a proof system (Cook & Reckhow, 1979).

from any of the axioms and that the problem has five (declarative) sentences.[6] We distinguish two forms of proof-tree shapes: We say that a proof tree is **linear** if each of its proof steps takes at most one premise that is not an axiom, and **nonlinear** otherwise.[7] In other words, every step in the solution to a linear problem uses only at most one fact not directly present in the problem; see Fig. 6 for an example. The proof tree given in Fig. 1 is *nonlinear* because the proof step that deduces the answer uses two premises that are not themselves axioms. Lastly, we define **canonical ordering** as the ordering of the leaf nodes when visited from left to right. Note that the sentences in the natural language annotation from Fig. 1 follow the canonical ordering of the proof tree. Informally, this is an easy way to order the sentences in the problem, since it matches the ordering of the proof steps. However, there exist multiple alternative orderings in general. In particular, given a proof tree that only contains commutative inference rules (see footnote 4), all orderings of the leaf nodes are valid. For instance, swapping the ordering of the first two leaf nodes in the tree in Fig. 1 yields a problem with the same proof.

## 4  EVALUATION FRAMEWORK

Equipped with the formalism explained in §3, we propose an evaluation framework that tests LLMs on MWPs that have arbitrarily complex proofs, called MathGAP. In §4.1, we explain the generation method behind MathGAP. We then outline how this method can be used for OOD evaluation in §4.2.

### 4.1  GENERATION METHOD

We propose a method for synthetic generation of problems and their CoT solution explanations. The approach consists of three high-level steps: (i) sample a proof tree, (ii) map the logical forms at the leaf nodes into a natural language problem, and (iii) map the proof steps into a CoT solution.[8] Fig. 1 gives an illustration of these steps; note that the agent properties are abbreviated for brevity.

In detail, the procedure is as follows: We first sample a logical form to label the root of the proof tree. We then sample an inference rule whose conclusion matches the root, and use it to connect the root node to child nodes representing the premises of the sampled inference rule. We repeat this procedure recursively for each premise at the leaf nodes of the current tree until a predetermined stopping criterion has been reached, resulting in a proof tree. The stopping criterion is user-specified; it could be, e.g., when the tree has reached a certain depth or width. The categorical properties for the logical forms (i.e., agent, entity, attribute, and unit) are sampled uniformly at random from some vocabularies.

We then convert each leaf node into a sentence via a natural language template in some order, forming the text corresponding to the body of the problem. The default order follows the proof tree's canonical ordering, but other orders are possible as well. The sentences are sampled uniformly at random from a predefined set of templates; some example templates are given in Table 1. The question is converted from the logical form at the root of the proof tree and is sampled from a set of interrogative sentences.

In the final step, we generate the CoT annotation for the problem. We first obtain a sequential ordering of the CoT proof steps by visiting the nodes in the proof tree with a post-order traversal.[9] We then convert each proof step into a sentence via a natural language template. The templates used for the axioms are identical to those used in the body of the problem text; that is, the proof steps corresponding to axioms simply repeat the relevant part of the problem text. For the other proof steps, we use templates designed to explain the computation step that is performed. Fig. 1

---

[6]In this paper, the width of the proof tree matches the number of declarative sentences (i.e., all sentences except the question) in the corresponding problem since we restrict the logical forms and sentences to be one-to-one. In general, this need not be the case since a single sentence could be written as a conjunction of several logical forms. One could also construct problems that have irrelevant sentences which either represent logical forms that will not be used in the proof or have no corresponding logical form at all.

[7]We note that our notion of linear proofs is distinct from linear logic (Girard, 1987).

[8]Our pipeline is similar to Opedal et al.'s (2024), except that we use a top-down procedure to generate full proof trees, rather than a sequential one to generate only axioms. This is a more general approach that enables generation of proofs with arbitrary shapes, e.g., nonlinear ones using `comp-eq`; see App. D. Our pipeline is also similar to the method proposed by Jin et al. (2023) for generating causal reasoning problems.

[9]Note that the choice of traversing the nodes in post order corresponds to applying each inference rule as soon as all of its premises are available. In addition, post-order traversal visits the axioms in canonical ordering. Alternative orders of tree traversal would violate at least one of these constraints.

illustrates the mapping between the nodes in the proof tree and the sentences in the CoT annotation. For instance, the third sentence in Fig. 1 explains the conclusion `cont(Lucy, 27, apple)`, which is deduced from the two leftmost nodes in the tree: "So Lucy has $17 + 10 = 27$ apples".

## 4.2 MathGAP: Mathematical Generalization on Arithmetic Proofs

With our generation method we can generate new synthetic problems where we control for the structure of the proof and its complexity. This mitigates the risk that the problems are contaminated. MathGAP can be used for various forms of systematic studies on arithmetic reasoning; here, we focus on in-context easy-to-hard OOD generalization. Specifically, we select a family of problems of interest (e.g., linear problems under `comp` inference rules) and generate a train-test split of problems in which problems in the test set are more complex than those in the training set (e.g., requiring deeper proofs).

The generation method can be flexibly adapted. One can add new logical forms and inference rules, restrict the choice of existing ones during sampling, or create logical forms for irrelevant sentences (Shi et al., 2023). One can also vary the text templates for the logical forms and inference rules, the vocabularies for agents, entities, attributes, and units, and the range of numbers from which the quantities are sampled. In addition, one may increase lexical and syntactic diversity by using an LLM to paraphrase the templated texts, e.g., through a procedure like Mishra et al.'s (2024). We do not perform such paraphrasing for the data in our experiments since it could introduce bias (Boguraev et al., 2024; Opedal et al., 2024) and produce problems that are unfaithful to their formal specifications.

## 5 Generalizing to Complex Proofs

We apply MathGAP to study how LLMs generalize from proof demonstrations of simple problems in context to solve problems of higher complexity at inference. We test OOD generalization through several sets of experiments: depth generalization for linear problems (§5.1), width generalization for linear problems (§5.2), depth generalization for nonlinear problems (§5.3), and generalization to permutations of the axioms (§5.4). Our public code repository provides further problem variations.

**Experimental setup.** For each set of experiments, the general experimental setup is as follows: We generate multiple test sets of different degrees of complexity with 400 problems in each. We then generate model predictions for the problems in these test sets under four different prompts: **(i)** A *zero-shot* baseline prompt, which contains only the test problem; **(ii)** A prompt of *primitive* examples, in which all problems contain only one proof step of the same inference rule that is used in the test problem; **(iii)** A prompt of examples within a *range* of complexities, all of which are simpler than the test problem; **(iv)** An *in-distribution* baseline prompt, containing examples that are of the same complexity as the test problem.

Note that prompts (i) through (iii) evaluate in-context OOD generalization, since the test examples are from a distribution that is different from that of the in-context examples. For prompts (ii) through (iv) we generate a new set of in-context examples for every test problem. This is done in order to avoid bias in the results towards any particular set of examples. We set the number of examples to 12, except for the experiments on nonlinear problems for which we use 5—see App. C for the rationale. Importantly, the in-context problems are provided together with their CoT solution annotations, as generated through the MathGAP pipeline. For prompt (iii), we make sure that there is at least one problem of every complexity in the range. This prompt can be viewed as a form of in-context curriculum learning (Bengio et al., 2009),[10] except that we randomize the ordering of the examples rather than arranging them in increasing order of complexity. This is done since in-context orderings can have a large effect on performance (Lu et al., 2022), and we do not want our results to overfit to any particular ordering. More details on the data and prompt are given in App. C.

Responses are generated using greedy decoding and a maximum context length of 4,096 tokens. Model predictions are obtained by extracting the last number occurring in the model output. We report answer accuracies and compute 95% confidence intervals using bootstrap sampling (Efron, 1992). We evaluate the following models on all test sets: Mixtral-8x7B (Jiang et al., 2024a), Llama3 with 8B and 70B parameters (Llama Team, 2024), GPT-3.5 Turbo and GPT-4o (OpenAI, 2024).

---

[10]Related approaches have been explored in previous work (Li et al., 2022; Liu et al., 2024b).

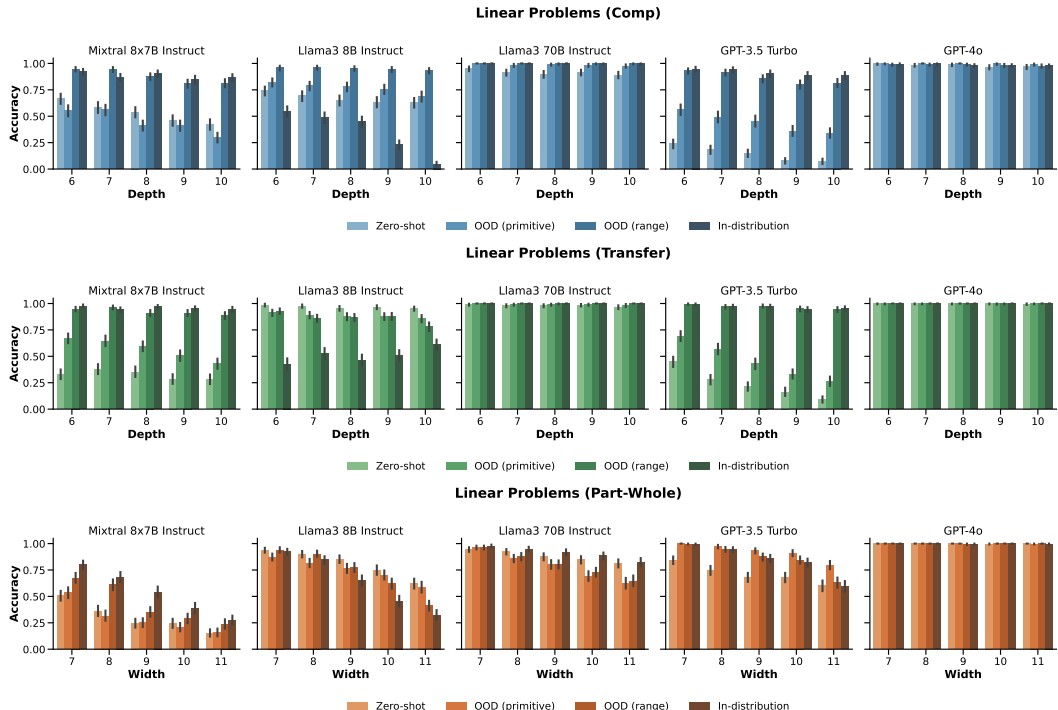

Figure 2: Answer accuracies for generalization to increasing depth and width for linear problems across models and in-context distributions. Depth is increased using inference rules involving `comp` and `transfer`. Width is increased using `partwhole`. See App. E for example problems.

## 5.1 LINEAR DEPTH GENERALIZATION

**Problem sets.** We consider linear problems using both `comp` and `transfer`. The five test sets consist of problems with depths between $6-10$. As such, the examples provided in the range of setting (iii) described above have depths between $1-5$. We give an example of a problem with depth 6 in App. E.1.

**Results.** The results are illustrated in the top and middle rows of Fig. 2. As expected, we observe a decreasing trend in performance as depth increases, for all in-context distributions. Curiously, GPT-3.5-Turbo is the only model for which adding in-context examples consistently leads to improvements. For the majority of cases, performance tends to be larger when providing a range of examples of varying complexity as compared to only primitive examples. The ranged OOD cases often show comparable performance to that of the in-distribution context, suggesting that these models benefit from demonstrations of problems with varying complexity. Llama3-8B is an outlier in that its solving accuracy for in-distribution context is (substantially) lower as compared to other prompts. When inspecting the outputs we see that the model often fails to answer the inference question, recognizing only that it has been provided a sequence of in-context examples. This could potentially be a consequence of how Llama3 is trained, using smaller context windows for the initial stages of pretraining (Llama Team, 2024). This appears to be remedied by scale since Llama3 70B shows near perfect performance.

## 5.2 LINEAR WIDTH GENERALIZATION

**Problem sets.** We test generalization to larger proof width using `partwhole` problems with a fixed depth of $1$. The five test sets have widths between $7-11$ and the problems in setting (iii) described above have widths between $2-6$. Note that this matches the width for the problems in §5.1; for those problems the width was equal to the depth plus one. App. E.2 gives an example problem.

**Results.** The results are shown in the bottom row of Fig. 2. Apart from the GPT models, the performance is generally lower as compared to deep problems with the same number of sentences (§5.1), often with a sharper decreasing trend. Such is the case even for Llama3 70B, which performed almost perfectly in the linear depth setting. This suggests that it is more difficult to generalize in terms of width than in terms of depth, which is somewhat surprising considering that these problems only have

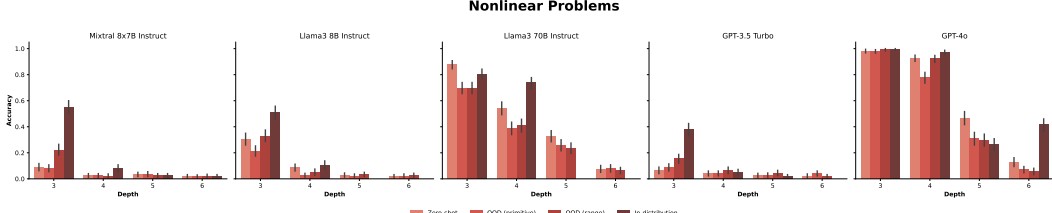

Figure 3: Answer accuracies for generalization to increasing depth and width for nonlinear problems across models and in-context distributions. Depth is increased using inference rules involving `comp` and `comp-eq`. The in-distribution contexts were too large to fit in context for the Llama3 models and GPT-3.5 Turbo at some of the higher depths. App. F shows results on o1-preview and DeepSeek-R1.

one proof step. For GPT-3.5 and Mixtral-8x7B, we observe that providing in-context examples improves accuracy over zero-shot in most cases. As for linear depth generalization, it seems that having a range of OOD examples of varying complexity is superior to only including primitive examples. For the Llama models, adding OOD in-context examples often leads to lower performance over zero-shot.

## 5.3 NONLINEAR DEPTH GENERALIZATION

**Problem sets.** We generate test datasets containing nonlinear problems with `comp` and `comp-eq` which have a similar form as the problem in Fig. 1. The procedure is explained in App. D and a more complex, depth 3 example is given in App. E.3. We generate a test set for each depth between $3-6$. Thus, the ranged OOD context setting contains examples of depths between $1-2$.

**Results.** The results are shown in Fig. 3. All models exhibit a performance trend that tends to zero as depth increases, albeit at different rates. For Mixtral-8x7B, Llama3-8B, and GPT-3.5 Turbo, the performance decreases rapidly across all context modes. These models benefit from in-distribution contexts for low depths, and in some cases, from a range of OOD examples. Llama3-70B and GPT-4o are more robust to this type of OOD distribution shift,[11] but for the deepest problems their performance tends to zero as well. Thus, the models are able to generalize to some extent but the error rate increases with complexity, suggesting that LLMs are noisy reasoners (Prabhakar et al., 2024). Moreover, it is worth noting that, for many cases, providing in-context examples does not significantly improve performance over zero-shot. On the other hand, for lower depths, it is usually beneficial to provide in-distribution contexts. We perform an additional analysis on o1-preview and DeepSeek-R1 in App. F.

**Comparison to linear problems.** Nonlinear problems are more difficult than the linear ones, even when controlling for the number of axioms given in the problem (i.e., its width). To see this, note that a nonlinear problem with depth 3 has 10 axioms (see App. D). This is the same number of axioms as the linear comparison and transfer problems (§5.1) with depth 9 and the linear part-whole problems (§5.2) with width 10. Comparing the results across those problem sets, we observe considerably lower accuracies for the nonlinear problems in most cases. We believe there are two main explanations for this: (i) Solving nonlinear problems requires keeping intermediate conclusions in memory while proving other intermediate conclusions, before being able to use them for further proof steps. It is thus less predictable where the relevant tokens will be located within the context window as compared to linear problems, in which intermediate conclusions are used immediately in a new proof step. (ii) Nonlinear problems use the `comp-eq` rule, which is most likely less frequent in the training set. Inspecting the model outputs we find support for both of these explanations—errors show up both when a previously deduced conclusion is needed as well as when a `comp-eq` is required in a proof step.

## 5.4 ORDER GENERALIZATION

**Problem sets.** LLMs are known to be sensitive to changes in the order of axioms in logical and mathematical reasoning problems (Chen et al., 2024; Eisape et al., 2024). Here, we present a systematic analysis of their sensitivity to a particular form of order permutation. We consider linear, left-leaning

---

[11]Note that GPT-4o performs better on depth 6 problems than depth 5 problems under the in-distribution prompt. Although we provided the same model id as in the other cases—which according to the documentation points to the original "gpt-4o-2024-05-13" snapshot—this particular test was performed at a later point in time.

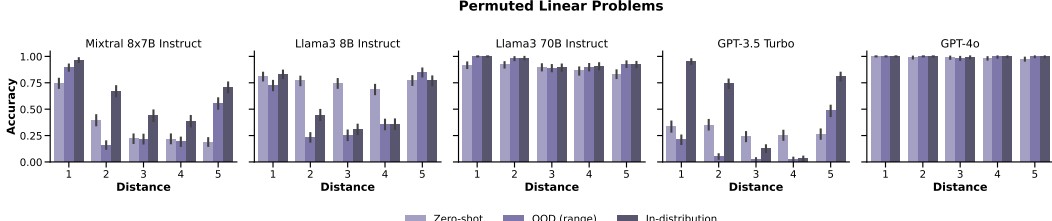

Figure 4: Answer accuracies for generalization to permutations across models and in-context distributions. We measure complexity as the distance of the movement of a sentence to the beginning of the problem, as compared to its position in the canonical ordering of the problem.

proof trees using `comp` that have depth 5, and generate problems that deviate from the canonical, left-to-right order of the leaf nodes. In particular, we pick one of the leaf nodes to visit first, and then visit the remaining leaf nodes in left-to-right order. This has the effect of moving one of the sentences in the canonically-ordered problem text to the beginning of the problem. An illustration is shown in Fig. 7 and App. E.4 gives further examples. We characterize complexity as the distance of the movement, with the hypothesis that a greater movement from a canonical ordering constitutes a more difficult problem. We create five test sets with movement distances between $1-5$. We do not include a primitive OOD in-context distribution since the notion of a primitive problem is ill-defined in this case. The range prompt includes movements of all distances between $1-5$ except the one present in the test set.

**Results.** The results are shown in Fig. 4. First, note that all models show lower performance on perturbed problems as compared to non-perturbed ones (compare to Fig. 2), suggesting that they are sensitive to simple perturbations of axiom orderings. Second, there is a nonlinear relationship between accuracy and distance of the movement, in contrast to the monotonically decreasing relation we expected. More specifically, the performance is the highest for the problems where the sentence is moved from near the beginning or the end of the problem.[12] In addition, including in-context examples gives a larger boost in performance for short and long movements, as compared to medium-length ones. Inspecting the model outputs, we observe that the models often make logical mistakes when the sentence that has been moved is needed for the subsequent proof steps. These kinds of mistakes appear to be more common than arithmetic errors.

## 6 CONCLUSION

This paper introduced MathGAP, a framework for evaluating LLMs on math word problems with proofs that are arbitrarily complex. MathGAP can flexibly generate synthetic arithmetic word problems with controllable proof tree characteristics. Importantly, this enables studies of easy-to-hard OOD generalization that avoid issues of data contamination, since a train-test split where the distributions are different can be created programmatically. In particular, the test set can be made arbitrarily more complex than the training set. We applied MathGAP to study whether LLMs can learn from simple examples in context to solve more complex problems in a test set. All models showed a decrease in performance as complexity of proof trees increases through depth and/or width. The LLMs struggled more with wider problems than with deeper problems, and were worse at solving problems with nonlinear shapes as compared to linear ones. We also found that LLMs are sensitive to the order in which sentences are presented. We further demonstrated that we can use MathGAP to construct problems that are challenging even for state-of-the-art reasoning models like OpenAI o1 and DeepSeek-R1—see App. F.

Code to generate problems with MathGAP, including problem types beyond those considered in this paper, can be found in our public code repository. Finally, see App. G for a discussion on limitations of the present work.

---

[12]This is consistent with findings on retrieval-augmented generation, for which models are better at using information that occurs near the beginning or end of the prompt (Liu et al., 2024a).

ACKNOWLEDGMENTS

We thank Alessandro Stolfo, Moritz Miller, Ryan Cotterell, Shehzaad Dhuliawala, Yanick Zengaffinen, and Yilmazcan Ozyurt for useful discussion. Andreas Opedal acknowledges funding from the Max Planck ETH Center for Learning Systems.

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

**Problem Text**

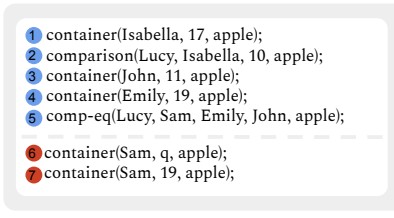

① Isabella has 17 apples.
② Lucy has 10 more apples than Isabella.
③ John has 11 apples.
④ Emily has 19 apples.
⑤ The number of apples that Lucy has more than Sam is the same as the difference between the number of apples that John has compared to Emily.

⑥ How many apples does Sam have?
⑦ Answer: 19

**World Model**

① container(Isabella, 17, apple);
② comparison(Lucy, Isabella, 10, apple);
③ container(John, 11, apple);
④ container(Emily, 19, apple);
⑤ comp-eq(Lucy, Sam, Emily, John, apple);

⑥ container(Sam, q, apple);
⑦ container(Sam, 19, apple);

Figure 5: World model (**right**) of a math word problem (**left**). Each sentence in the problem text is represented by a logical form which consists of a predicate with property arguments. The logical forms in the body are used as axioms in the proof of the problem, as shown in Fig. 1. The sentences and logical forms labeled (1) through (5) are the body of the problem, (6) is the question and (7) is the answer.

Mingyu Zong and Bhaskar Krishnamachari. Solving math word problems concerning systems of equations with GPT-3. *Proceedings of the AAAI Conference on Artificial Intelligence*, 37(13): 15972–15979, Sep. 2023. URL https://ojs.aaai.org/index.php/AAAI/article/view/26896.

## A   INTUITION ON LOGICAL FORMS

A `cont` logical form expresses the fact that an agent owns a particular entity in some quantity (e.g., *"Alice has 5 apples"*). A `comp` logical form expresses how much more some agent owns of a particular entity than some other agent (e.g., *"Bob has 3 fewer apples than Alice"*). A `transfer` logical form expresses the transfer of a certain quantity of an entity from one agent to another (e.g., *"Alice gave Bob 3 apples"*). A `partwhole` logical form expresses a superset relation between the combined entities of several agents (e.g., *Alice and Bob combine their fruits*). Finally, a `comp-eq` logical form expresses that the quantity of a particular `comp` between two agents is equal to the difference of the quantities of the entity of two other agents (e.g., *"The number of apples that Alice has more than Bob is the same as the difference between the number of apples that Charlie and David have"*).

Fig. 5 gives an example problem text along with its world model.

## B   DEFINING INFERENCE RULES WITH NATURAL DEDUCTION

Our inference rules can be written using the primitive inference rules present in natural deduction (Gentzen, 1935; Pfenning, 2004) as applied to first-order logic expressions; see Opedal et al. (2023, App. D) for details on how the formalism described in §3 is a form of first-order logic. In this section we illustrate an example using the inference rule

$$\frac{\texttt{cont(a, } q_1 \texttt{, e)} \quad \texttt{comp(b, a, } q_2 \texttt{, e)}}{\texttt{cont(b, } q_1 + q_2 \texttt{, e)}},$$

involving `comp`. This inference rule is composed of rules in natural deduction as follows:

1. $\exists v_1(\texttt{Owner}(v_1, \texttt{a}) \wedge \texttt{Measure}(v_1, q_1) \wedge \forall x \in v_1.\texttt{e}(x))$ by Axiom ($1^{st}$ premise, definition of container as in Opedal et al. 2023, App. D.2.1).

2. $\texttt{Owner}(v_1, \texttt{a}) \wedge \texttt{Measure}(v_1, q_1) \wedge \forall x \in v_1.\texttt{e}(x)$ by $\exists$-elimination [1].

3. $\exists e_1(\texttt{ComparisonAdd}(e_1) \wedge \texttt{Source}(e_1, v_1) \wedge \texttt{Target}(e_1, v_2) \wedge \texttt{Time}(v_1) = \texttt{Time}(v_2) \wedge \exists r(\texttt{Arg}(e_1, r) \wedge \texttt{Measure}(r, q_2) \wedge \forall x \in r.\texttt{e}(x)))$ by Axiom ($2^{nd}$ premise, definition of comparison as in Opedal et al. 2023, App. D.2.2).

4. $\texttt{ComparisonAdd}(e_1) \wedge \texttt{Source}(e_1, v_1) \wedge \texttt{Target}(e_1, v_2) \wedge \texttt{Time}(v_1) = \texttt{Time}(v_2) \wedge \exists r(\texttt{Arg}(e_1, r) \wedge \texttt{Measure}(r, q_2) \wedge \forall x \in r.\texttt{e}(x))$ by $\exists$-elimination [3].

5. $\exists r(\text{Arg}(e_1, r) \wedge \text{Measure}(r, \mathrm{q}_2) \wedge \forall x \in r.\text{e}(x))$ by $\wedge$-elimination [4].

6. $\text{Arg}(e_1, r) \wedge \text{Measure}(r, \mathrm{q}_2) \wedge \forall x \in r.\text{e}(x)$ by $\exists$-elimination [5].

7. $\text{Arg}(e_1, r) \wedge \forall x \in r.\text{e}(x)$ by $\wedge$-elimination [6].

8. $\text{ComparisonAdd}(e_1) \wedge \text{Arg}(e_1, r) \wedge \forall x \in r.\text{e}(x)$ by $\wedge$-introduction [7].

9. $\forall e \forall r(\text{ComparisonAdd}(e) \wedge \text{Arg}(e, r) \wedge \forall x \in r.\text{e}(x) \rightarrow \exists v \exists o \exists q(\text{Target}(e, v) \wedge \text{Owner}(v, o) \wedge \text{Measure}(v, q) \wedge \forall x \in v.\text{e}(x)))$ by Axiom (theorem on the existence of containers from existing relations as in Opedal et al. 2023, App. D.4).

10. $\text{ComparisonAdd}(e_1) \wedge \text{Arg}(e_1, r) \wedge \forall x \in r.\text{e}(x) \rightarrow \exists v \exists o \exists q(\text{Target}(e_1, v) \wedge \text{Owner}(v, o) \wedge \text{Measure}(v, q) \wedge \forall x \in v.\text{e}(x)))$ by $\forall$-elimination [9].

11. $\exists v \exists o \exists q(\text{Target}(e_1, v) \wedge \text{Owner}(v, o) \wedge \text{Measure}(v, q) \wedge \forall x \in v.\text{e}(x))$ by $\rightarrow$-elimination [8, 10].

12. $\text{Target}(e_1, v_2) \wedge \text{Owner}(v_2, \mathrm{b}) \wedge \text{Measure}(v_2, \mathrm{q}_t) \wedge \forall x \in v_2.\text{e}(x)$ by $\exists$-elimination [11].

13. $\text{Owner}(v_2, \mathrm{b}) \wedge \text{Measure}(v_2, \mathrm{q}_t) \wedge \forall x \in v_2.\text{e}(x)$ by $\wedge$-elimination [12].

14. $\forall e \forall v_s \forall v_t \forall m_s \forall m_t \forall r(\text{ComparisonAdd}(e) \wedge \text{Arg}(e, r) \wedge \text{Source}(e, v_s) \wedge \text{Target}(e, v_t) \wedge \text{Measure}(v_s, m_s) \wedge \text{Measure}(v_t, m_t) \rightarrow m_s + r = m_t)$ by Axiom (theorem defining the semantics of comparison as in Opedal et al. 2023, App. D.3).

15. $\text{ComparisonAdd}(e_1) \wedge \text{Arg}(e_1, r) \wedge \text{Source}(e_1, v_1) \wedge \text{Target}(e_1, v_2) \wedge \text{Measure}(v_1, \mathrm{q}_1) \wedge \text{Measure}(v_2, \mathrm{q}_t) \rightarrow \mathrm{q}_1 + \mathrm{q}_2 = \mathrm{q}_t$ by $\forall$-elimination [14].

16. $\text{Measure}(v_1, \mathrm{q}_1)$ by $\wedge$-elimination [2].

17. $\text{Measure}(v_2, \mathrm{q}_t)$ by $\wedge$-elimination [13].

18. $\text{ComparisonAdd}(e_1) \wedge \text{Arg}(e_1, r)$ by $\wedge$-elimination [8].

19. $\text{Source}(e_1, v_1) \wedge \text{Target}(e_1, v_2)$ by $\wedge$-elimination [4].

20. $\text{ComparisonAdd}(e_1) \wedge \text{Arg}(e_1, r) \wedge \text{Source}(e_1, v_1) \wedge \text{Target}(e_1, v_2) \wedge \text{Measure}(v_1, \mathrm{q}_1) \wedge \text{Measure}(v_2, \mathrm{q}_t)$ by $\wedge$-introduction [4, 16, 17, 18, 19].

21. $\mathrm{q}_1 + \mathrm{q}_2 = \mathrm{q}_t$ by $\rightarrow$-elimination [15, 20].

22. $\text{Owner}(v_2, \mathrm{b}) \wedge \text{Measure}(v_2, \mathrm{q}_1 + \mathrm{q}_2) \wedge \forall x \in v_2.\text{e}(x)$ by $=$-elimination [13, 21].

23. $\exists v_2(\text{Owner}(v_2, \mathrm{b}) \wedge \text{Measure}(v_2, \mathrm{q}_1 + \mathrm{q}_2) \wedge \forall x \in v_2.\text{e}(x))$ by $\exists$-introduction [22].

Note that the conclusion (step 22) is the first-order logic equivalent of `cont(b, q₁ + q₂, e)`, which is the conclusion of the comparison inference rule.

## C  MORE DETAILS ON EXPERIMENTS

**Generated problems.** The quantities are sampled uniformly at random from the range $2-20$. We use a handwritten list of 52 English-language names for most problems, with the exception of the larger nonlinear problems for which we use a larger list of 4,945 predominantly English-language names from https://github.com/dominictarr/random-name. For the entities we use a handwritten word list with 51 items. We restrict each problem to have one entity, We determine uniformly at random whether the problem has attributes, units, or neither of them.

**Prompts.** We use 12 in-context examples for all test sets except for the nonlinear problems, for which we use 5. We use fewer in-context examples for nonlinear problems since the number of sentences in a nonlinear problem increases exponentially with depth. The number 12 was deemed large enough so that additional examples would have a negligible positive impact on performance; see, e.g., Agarwal et al. (2024, Fig. 7). Moreover, in preliminary experiments we found that the number of in-context examples had little effect on performance. The in-context examples are written out using the pattern "*Q:* `{problem}`\n*A:* `{CoT solution}`\n".

**Models.**  The maximum context windows for Mixtral-8x7B, Llama3 8B, Llama3 70B, GPT-3.5 Turbo and GPT-4o are 65k, 8k, 8k, 16k, 128k, respectively. Combined with a maximum number of output tokens of 4,096, the models have a theoretically sufficient context size to solve all test problems. The largest test problems (nonlinear depth 6) have around 2.5k tokens, including CoT annotation.

For GPT-3.5 Turbo and GPT-4o we used "gpt-3.5-turbo-0125" and "gpt-4o-2024-05-13" as the model ids for all experiments.

## D  NONLINEAR COMPARISON PROBLEMS

In this section we explain why we have introduced the `comp-eq` predicate to generate nonlinear trees. We also give the procedure for generating the nonlinear problems used for the experiments in §5.3.

**Motivating example.**  We first motivate why we introduce the `comp-eq` predicate to build proof trees with nonlinear shape of arbitrary depth. Consider the following problem which is based solely on logical forms with `cont` and `comp`: *"Alice has 5 apples. Bob has 3 more apples than Alice. Charlie has 10 apples. David has 2 more apples than Charlie. How many more apples does David have than Bob?"*. This problem is represented by the following nonlinear proof tree of depth 2:

$$\frac{\dfrac{\texttt{cont(A, 5, a)}\quad\texttt{comp(B, A, 3, a)}}{\texttt{cont(B, 8, a)}}\quad\dfrac{\texttt{cont(C, 10, a)}\quad\texttt{comp(D, C, 2, a)}}{\texttt{cont(D, 12, a)}}}{\texttt{comp(D, B, 4, a)}}.$$

Now, say we are following the top-down generation procedure described in §4.1 and wish to expand the tree to higher depths. However, the two leaf nodes labeled with `comp` are impossible to expand further under the given inference rules. For instance, the `comp(B, A, 3, a)` would be deduced by the two premises `cont(A, 5, a)` and `cont(B, 8, a)`, but those two logical forms are already present elsewhere in the tree and are thus not available. This leaves only the two `cont` nodes for further expansion. However, any further subtrees of the `cont` nodes will be linear, since none of the `comp` nodes generated along the way will be able to be expanded.[13]

The inference rule with `comp-eq` addresses this issue. To see this, replace the subtree rooted at `cont(B, 8, a)` in the proof tree above with some subtree that uses an inference rule with `comp-eq`, e.g.,

$$\frac{\texttt{cont(A, 5, a)}\quad\texttt{comp(E, F, 4, a)}\quad\texttt{comp-eq(B, A, E, F)}}{\texttt{cont(B, 8, a)}}.$$

Note that, in this case, both `cont(A, 5, a)` and `comp(E, F, 4, a)` can be expanded. In particular, `comp(E, F, 4, a)` introduces two new agents `E` and `F`, so it can be expanded similarly to the root node from our example above.

**Procedure.**  We want to sample a nonlinear proof tree with depth $D$. We start by sampling either a `cont` node or a `comp` node uniformly at random to determine the root of the tree. Next, we sample an inference rule for which the root logical form is the conclusion. We then sample inference rules from the child nodes in breadth-first order until all leaves except the ones labeled with `comp-eq` (which cannot be expanded) have depth $D$. For `comp` nodes, there is only one inference rule possible (the third row in Table 2). For `cont` nodes we sample a `comp` rule (i.e., the first row in Table 2) if the `cont` node has depth $D - 1$, and a `comp-eq` rule (i.e., the fifth row in Table 2) otherwise. This ensures that we can continue to grow the tree nonlinearly until the specified depth, following the reasoning given in the paragraph above.

Note that the width of the proof trees generated with this procedure will be the number of leaf nodes in a perfect binary tree with depth $D$, i.e., $2^D$, plus the number of nodes labeled with `comp-eq`. Thus, a depth 2 problem will have a width of either 4 or 5, a depth 3 problem will have a width of 10,

---

[13]This reasoning assumes that we only use the inference rules with `cont` and `comp` from Table 2. Deeper nonlinear problems could be achieved by using the `partwhole` rule as well.

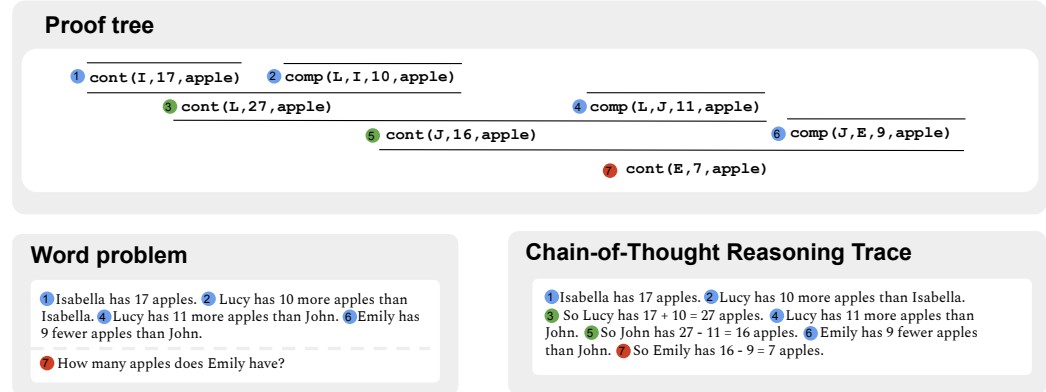

Figure 6: Example of a linear problem using `comp` with depth 3 and width 4. This problem is linear because each parent node in the tree has at most one child that is not a leaf node. Compare to the nonlinear problem shown in Fig. 1.

and a depth 4 problem will have a width of either 20 or 21. In general, the width $W(D)$ for a proof tree of depth $D$ is:

$$W(D) = 2^D + \sum_{d=0}^{D-2} f_c(d),$$

where $f_c(d)$ gives the number of `comp-eq` nodes as a function of the level $d$, following the recursion

$$f_c(0) = \begin{cases} 1 & \text{if root is } \texttt{cont} \\ 0 & \text{if root is } \texttt{comp} \end{cases}$$

$$f_c(1) = \begin{cases} 1 & \text{if root is } \texttt{cont} \\ 2 & \text{if root is } \texttt{comp} \end{cases}$$

$$f_c(d) = f_c(d-1) + 2f_c(d-2), d \geq 2.$$

## E    EXAMPLE PROBLEMS AND ILLUSTRATIONS

In this section we provide example problems for each set of experiments (§§ 5.1 to 5.4).

### E.1    EXAMPLE LINEAR COMPARISON PROBLEM (DEPTH 6)

*Jackson has 16 red bottles of soap. Jackson has 10 more red bottles of soap than Abigail. Joseph has 18 more red bottles of soap than Abigail. Joseph has 14 fewer red bottles of soap than James. Michael has 2 more red bottles of soap than James. Ryan has 16 fewer red bottles of soap than Michael. Mia has 10 more red bottles of soap than Ryan. What is the number of red bottles of soap that Mia has?*

### E.2    EXAMPLE PARTWHOLE WIDTH PROBLEM (WIDTH 7)

*Emily has 5 apples. Lily has 8 bananas. Abigail has 9 bananas. Benjamin has 11 grapes. Christopher has 20 apples. Mila has 16 grapes. Sophia has 11 watermelons. If everyone sums up the fruits that they have, how many fruits does everybody have in total?*

### E.3    EXAMPLE NONLINEAR COMPARISON PROBLEM (DEPTH 3)

*Ella has 11 yellow plates. Ella has 19 fewer yellow plates than Jacob. Evelyn has 16 yellow plates. Daniel has 10 yellow plates. The number of yellow plates that Emma has more than Jacob is the same as the difference between the number of yellow plates that Evelyn has compared to Daniel. Lucy has 2 yellow plates. Amelia has 6 more yellow plates than Lucy. Layla has 17 yellow plates.*

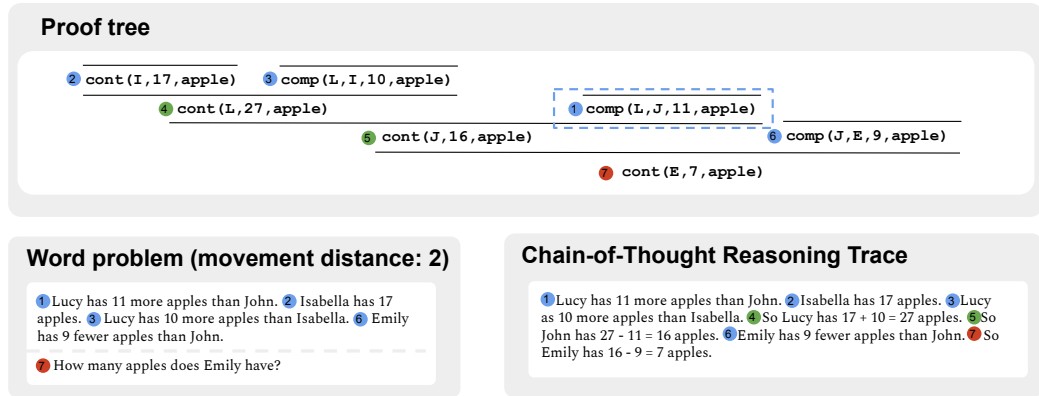

Figure 7: The same proof tree as Fig. 6 but with a permuted ordering as in the experiments discussed in §5.4. In this specific example, the leaf node labeled with `comp(L, J, 11, apple)` is visited first (the other nodes follow post-order traversal). This translates into moving the sentence "Lucy has 11 more apples than John" to the front of the problem. As compared to the canonically ordered problem shown in Fig. 6, the sentence has moved two steps.

*Layla has 13 fewer yellow plates than Sophia. The number of yellow plates that Emma has more than Henry is the same as the difference between the number of yellow plates that Amelia has compared to Sophia. What is the number of yellow plates that Henry has?*

### E.4 PERMUTED PROBLEMS

#### CANONICALLY ORDERED PROBLEM TEXT

*Nicholas has 19 computers. Lucy has 6 fewer computers than Nicholas. Harper has 6 fewer computers than Lucy. John has 10 more computers than Harper. Abigail has 15 fewer computers than John. Abigail has 12 fewer computers than Logan. How many computers does Logan have in their collection?*

#### PERMUTED PROBLEM TEXT (DISTANCE 1)

***Lucy has 6 fewer computers than Nicholas.*** *Nicholas has 19 computers. [Original Position] Harper has 6 fewer computers than Lucy. John has 10 more computers than Harper. Abigail has 15 fewer computers than John. Abigail has 12 fewer computers than Logan. How many computers does Logan have in their collection?*

#### PERMUTED PROBLEM TEXT (DISTANCE 3)

***John has 10 more computers than Harper.*** *Nicholas has 19 computers. Lucy has 6 fewer computers than Nicholas. Harper has 6 fewer computers than Lucy. [Original Position] Abigail has 15 fewer computers than John. Abigail has 12 fewer computers than Logan. How many computers does Logan have in their collection?*

#### PERMUTED PROBLEM TEXT (DISTANCE 5)

***Abigail has 12 fewer computers than Logan.*** *Nicholas has 19 computers. Lucy has 6 fewer computers than Nicholas. Harper has 6 fewer computers than Lucy. John has 10 more computers than Harper. Abigail has 15 fewer computers than John. [Original Position] How many computers does Logan have in their collection?*

Figure 8: Answer accuracies for generalization to increasing depth and width for nonlinear problems for OpenAI o1-preview and DeepSeek-R1 under zero-shot prompting and a maximum context length of 4,096 tokens.

## F    ADDITIONAL ANALYSIS ON REASONING MODELS

Fig. 8 shows additional results on the OpenAI o1-preview model ("o1-preview-2024-09-12") and DeepSeek-R1 (DeepSeek-AI, 2025), evaluated on the nonlinear test sets. We only used zero-shot prompting due to the high costs of performing inference on these models. Using a simple, straightforward prompt is also consistent with OpenAI's current recommendations. We see that these models show superior performance to all models from the main text, with o1 generalizing better than R1. However, as with the other models, the performance decreases as complexity increases.

Importantly, we noted that performance can be drastically improved by increasing the limit on the number of output tokens. The results presented in Fig. 8 used a token limit of 4,096, like the experiments in the main paper. For o1, we compared the effect of the token limit on an additional test set of 400 nonlinear problems with depth 7. With a token limit of 4,096, the model could only answer $0.25\%$ of the problems correctly. However, with a token limit of 10,000, it answered $76.5\%$ of the problems correctly. For R1, the performance on problems with depth 6 is increased to $77.5\%$ when using a token limit of 10,000. These results are noteworthy and suggests that the inference techniques used by these models is highly effective for decomposing complex problems into smaller steps.

To test the limits of these models, we performed a final evaluation on a subset of 100 of the depth 7 problems where the sentences in each problem were *randomly ordered* (apart from the question). Recall from §3.2 that permuting the sentences in random order is valid since all the inference rules used in our nonlinear problems are commutative. In this experiment, we allowed 25,000 output tokens for both models, which is the current recommendation by OpenAI. On this set, the answer accuracy was only $5.0\%$ for o1 and $11.0\%$ for R1. Thus, we conclude that MathGAP is future-proof in the sense that it can generate problems on which current state-of-the-art reasoning models fail. We stress, however, that the aim of this paper is not to construct a challenging test set for state-of-the-art models; there are several other ways that MathGAP can be used to create problems that might be even more difficult.

## G    LIMITATIONS

We did not prioritize linguistic diversity in our problem sets; future work may investigate how our findings generalize to distributions of problems with higher linguistic diversity. We did also not consider or study the possibility of token biases (Jiang et al., 2024b). It is beyond the scope of the present study to consider problem texts in languages other than English; however, it is straightforward to extend MathGAP to other languages. In addition, we cannot guarantee that the distributions of our generated problems are different from those of any potential internal data-generating mechanisms from Meta or OpenAI. However, we deem it to be unlikely that they use a similar formalism as we do here, and it gets increasingly unlikely that problems are similar as complexity increases.

While the more complex nonlinear problems are indeed challenging even for the most capable models, we note the purpose of our study was not to create a challenging evaluation set for state-of-the-art LLMs. If that is the goal, one could create harder problems by combining several inference rules (e.g., additionally including the rate concept; Opedal et al., 2023), allowing a wider variety of proof-tree shapes, and presenting the axioms in arbitrary orderings. It is also possible to create more syntactically complex templates (and paraphrase those using LLMs) and use numbers on which it is more challenging for models to perform arithmetic (Razeghi et al., 2022).

Finally, it is hard to make a direct comparison between the difficulty of linear and nonlinear problems, since they use different inference rules. In particular, nonlinear proofs contain `comp-eq` predicates, which are not present in linear proofs.

