# OpenReview forum: "MathGAP: Out-of-Distribution Evaluation on Problems with Arbitrarily Complex Proofs"
_ICLR.cc/2025/Conference — ICLR 2025 Poster_

### Official Review · Reviewer_iPba · 2024-10-28

**Soundness:** 3
**Presentation:** 4
**Contribution:** 2
**Rating:** 6
**Confidence:** 3

**Summary:**

This paper proposes a math proof benchmark, called MATHGap, by constructing proof trees with predefined predicates and inference rules. Evaluation on a number of LLMs shows that LLM's performance decreases as the proof complexity increases. Many LLMs do not do well in these problems.

**Strengths:**

1. The authors did a good job elaborating the method. Section 3 provides a good background for readers to understand the method.
2. Constructing out-of-domain math problems using the logical form is a good way to evaluate the math capabilities of LLMs.

**Weaknesses:**

1. Despite the many LLMs being tested, the paper seems to miss the latest state-of-the-art LLMs, including Llama 3.1, Llama 3.2 (since Llama 3.2 is only recently released, I understand that the authors did not have time to include it), and GPT-O1. This is rather important, because the linear problems were pretty well-resolved by GPT-4O, and Llama 3 is also already very competitive. Including the more recent models might change the conclusions. Therefore, the current conclusion seems a bit outdated.

2. The authors seem to relate the failure of LLM's performance on the more complicated proofs to the OOD generalization failures. However, this is not necessarily the case because it may as well just be the performance degradation due to the problems being more challenging. In many cases, the in-distribution examples do not improve the performance over OOD examples, which is evidence that the performance drop is due to increased difficulty, not the generalization gap.

3. The analysis in the paper looks a bit shallow. It does not show much insight into why and how the LLMs fail. For example, is the performance degradation as complexity increases simply due to the more proof steps? Namely, it may as well be the case that the error rate of each step is constant; it is just the increased number of steps that drive down the overall success rate. In this case, the model scales fine and it is just the success criterion that becomes more stringent. Also, what are the typical failure modes? Are the errors simple arithmetic errors, reasoning errors, or errors in parsing the final results? Why does the in-context example fail to improve the performance? Why is there a nonlinear relationship between the distance of the ordering and the performance? It would make the paper stronger if the authors could perform more experiments to answer these research questions.

4. The MATHGap dataset looks too constrained for me. It only contains 5 predicates and 5 inference rules. As a result, the linear problems seem too easy for the latest state-of-the-art models. It would be nice to study how the performance scales with the number of predicates and inference rules, and whether the LLMs can retrieve the correct inference rules where there are many.

**Questions:**

I would appreciate if the authors could investigate the research questions raised in the 'weaknesses' section. In addition -

Why were different predicates used in different parts of the analysis, i.e. comp and transfer for depth analysis, part-whole for width analysis, etc.?

---

> ### Author Response · Authors · 2024-11-22
>
> Thank you for your review and detailed feedback. We are glad that you agree that our method is a good way of evaluating math capabilities of LLMs. We address your suggested weaknesses below:
>
> 1. [See responses below]
>
> > Despite the many LLMs being tested, the paper seems to miss the latest state-of-the-art LLMs, including [...] GPT-O1
>
> We agree that including o1 is interesting so we have added some experiments to the paper. Please refer to App. C and references in the intro, conclusion, and 5.3. Some findings are: (i) we see a downward trend for nonlinear problems, (ii) but the performance depends on the number of output tokens we allow and (iii) it is highly sensitive to random orderings. See the updated version of the paper for more details – hope you find the results interesting.
>
> > [...] Llama 3 is also already very competitive. Including the more recent models might change the conclusions. Therefore, the current conclusion seems a bit outdated.
>
> Regarding that Llama3 70B already performs well on linear problems: We think there is a deeper point here beyond “what is model X’s performance on complexity Y”. The main difference between Llama3 8B and 70B is scale, and not training method or data. So if we believe that scale does not alone lead to perfect performance, we can conclude that there exists a set of more complex linear problems that the 70B model would fail on as well. The appeal of MathGAP is that you can generate such examples synthetically. We did not do so here particularly for Llama 70B and linear examples, since our primary aim was not to find the performance frontier of SoTA models. (Although as we note in our limitations section, our framework could be used in such a manner if desired.) Moreover, note that all models fail on complex nonlinear problems, demonstrating that MathGAP is future proof as an evaluation framework for capable models.
>
> 2. That is a good point, and is partly why we tried to be careful in our claims on generalization. We added a sentence on this in the introduction. We are happy to make further edits if there are specific parts of the text where our interpretations of the results are misleading, please feel free to point us to any such instance.
>
> 3. You raise a lot of interesting questions. That is great, because it is precisely what we are hoping to elicit with our evaluation framework. To answer your first set of questions would require control of the training/finetuning data, which is made possible for future studies using MathGAP. For some of the others we have added discussion to the text. For instance, there are more logical errors than arithmetic ones (see end of 5.4) and the quadratic relationship might be related to similar findings for RAG (see footnote 9).
>
> 4. [See responses below]
>
> > The MATHGap dataset looks too constrained for me.
>
> The predicates follow a standard taxonomy for grade school math problems (Riley et al., 1983). In addition, there exists a rate concept (e.g., “there are 5 apples per basket”) which was not included in our analysis but will be added to the code. Moreover, the inference rules can be combined and instantiated in several interesting ways, most of which we did not consider in our experiments for this paper.
>
> > the linear problems seem too easy for the latest state-of-the-art models.
>
> We disagree with the position that it is a weakness that some of the models do well on one particular type of problems, namely, linear ones. Instead, we view it as an interesting result which suggests that the models are able to do some form of generalization (we assume that higher depth linear problems are not very prevalent in the dataset but that would of course need to be verified to draw any conclusions).
>
> > Why were different predicates used in different parts of the analysis, i.e. comp and transfer for depth analysis, part-whole for width analysis, etc.?
>
> The reason we have separate datasets for comp and transfer is that we wanted to see if there is any significant difference in performance, which there indeed turns out to be for Mixtral and Llama 8B. We use part-whole for the width experiments since the part-whole inference rule is the only one that supports an arbitrary number of premises – see Table 2.

---

> > ### Comment · Reviewer_iPba · 2024-11-24
> > **Thanks for the response**
> >
> > I want to thank the authors for their response. I adjusted my score accordingly. Still, the in-depth analysis in the paper is sporadic and dispersed. It would be better if the paper had a more structured in-depth analysis.

---

### Official Review · Reviewer_d9We · 2024-11-01

**Soundness:** 3
**Presentation:** 3
**Contribution:** 3
**Rating:** 6
**Confidence:** 3

**Summary:**

Paper presents a framework for systematically studying the OOD reasoning capabilities of LLMs by programmatically generating training and test examples. Paper also presents analysis of how existing models behave for MathGAP problems under different settings.

**Strengths:**

Originality
- Current LLM reasoning benchmarks are mostly comprised of fixed datasets, which might be susceptible to data leakage and do not offer a way change different characteristics of the datasets such as difficulty
- MathGAP offers a way to systematically change the different aspects of the problem, size as depth, width and (non)linearity of problem tree, and constructs new problems every time, allowing for more controlled experiments on OOD generalization

Quality/Clarity
- Paper is well written and easy to understand

Significance
- in addition to providing a new benchmark to study generalization, paper also conducts analysis of current LLMs behavior using the proposed benchmark
- Presents interesting findings such as some models performing better when given OOD in-context examples rather than IID in-context examples

**Weaknesses:**

MathGAP examples seem to mostly involve a certain type of arithmetic problem, it would be interesting to expand to scope to include other types of reasoning problems

It would also be interesting to also study the finetuning generalization behavior of LLMs

**Questions:**

See weaknesses

---

> ### Author Response · Authors · 2024-11-22
>
> Thank you for the positive review and feedback. We are pleased that you found the findings to be interesting.
>
> > MathGAP examples seem to mostly involve a certain type of arithmetic problem, it would be interesting to expand to scope to include other types of reasoning problems
>
> Regarding the scope, we agree that it would be useful if our method was to be generalized to other kinds of reasoning problems as well. While our framework does not cover all types of arithmetic word problems, it follows a taxonomy that is commonly used in the learning sciences (Riley et al., 1983) and should thus have a good coverage of standard grade school math problems. We considered a particular family of interesting proof tree structures for this work; we agree that there exist other structures that could be analyzed, and our tool allows researchers to do this.
>
> > It would also be interesting to also study the finetuning generalization behavior of LLMs
>
> We also agree that it is interesting to analyze how generalization behavior interacts with pretraining and finetuning in various ways. While beyond the scope of this paper, MathGAP allows for such analyses and it is something we are planning for the future.

---

> > ### Comment · Reviewer_d9We · 2024-11-25
> >
> > Thanks for the response. I will maintain my score.

---

### Official Review · Reviewer_tyGU · 2024-11-02

**Soundness:** 3
**Presentation:** 4
**Contribution:** 3
**Rating:** 8
**Confidence:** 5

**Summary:**

The paper introduces MathGAP, a framework for evaluating LLMs' mathematical abilities. The paper notes two main motivations for creating MathGAP: 1) data leakage/contamination and 2) lack of comprehensive and systematic evaluation signals (e.g., OOD generalization, different complexity levels, etc.). The authors show that the performance of almost all LLMs drops as the difficulty (measured by depth/width of proof tree) increases. Moreover, the paper shows that LLMs are sensitive to the order in which input assumptions/axioms are presented, suggesting limitations in the perceived reasoning capabilities of LLMs.

**Strengths:**

1- Given the unfortunate hype over LLMs, I believe it is crucial to have objective and solid evaluation metrics for LLMs and separate science from hype. This is exactly what this work is aiming to do.
2- The paper is extremely well written and the authors present the results very well.
3- Overall, the experiments are well-designed. However, they could have been extended to provide more insights (see below).

**Weaknesses:**

1- It is not clear how much "new insight" this work provides as many of the results have already been reported in other similar contexts. For instance, Dziri et al. also has a similar notion of graph depth/width, and shows that by increasing depth and width for both ID/OOD setups the performance drops significantly (see Fig. 5 of [1] for example). However, some open questions have not been addressed yet. For instance, from the results, it is not clear why the Llama3-70B model has a near-perfect score on "linear depth" scenarios: is it because of its training data, scale, or something else? Moreover, why is the performance gap between linear/non-linear benchmarks so significant? Do we know why the LLMs struggle so much in the nonlinear problems (e.g., Figure 3 of the paper) even when depth is small?

2- I believe given the nature of MathGAP, the paper could have explored more fine-grained evaluation scenarios beyond the final accuracy with in-context prompts. For instance, according to the note in section 5, the number of shots is fixed to 12 and the footnote mentions that this is large enough according to another work. However, the impact of context on model performance on MathGAP is unclear: Does the number of shots have a similar impact on linear/non-linear setups? Why?

3- The paper does not mention how large the overall set of proper names and values were. For instance, we know that LLMs have token bias and this can have an impact on results [2]. In addition, it is not clear what range of numbers was used and how many of the mistakes were arithmetic mistakes versus logical mistakes.

4- The logical forms and inference rules used are relatively simple and may not capture more complex mathematical relationships. However, this is not a critical issue compared to other issues as we know that models still struggle in these simple cases as well.

Overall, I think the work is borderline in its current form. I believe this work has potential to have a significant contribution if it can provide more insights and explanations that lead to better understanding of LLMs.

[1] Dziri, Nouha, et al. "Faith and fate: Limits of transformers on compositionality." Advances in Neural Information Processing Systems 36 (2024).
[2] Jiang, Bowen, et al. "A Peek into Token Bias: Large Language Models Are Not Yet Genuine Reasoners." arXiv preprint arXiv:2406.11050 (2024).

**Questions:**

See the questions from my comments above.

---

> ### Author Response · Authors · 2024-11-22
>
> Thank you for the positive, encouraging comments and feedback. We are pleased that you appreciate this line of work and that you found our draft to be well written. We respond to the weaknesses below:
>
> 1. [See responses below]
>
> > For instance, Dziri et al. also has a similar notion of graph depth/width, and shows that by increasing depth and width for both ID/OOD setups the performance drops significantly (see Fig. 5 of [1] for example)
>
> Dziri et al.’s analysis is interesting as well but their computation graphs are different. Most notably, their nodes are labeled with real values, while ours are labeled with richer annotations about semantics (logical forms). Thus, our framework enables more granular analysis. For example, we can distinguish between comparison type problems and transfer type problems that have the same computation graphs. The way they define width is also different. As a consequence, even though the results point in similar directions, the implications are different. Our approach is less tied to specific implementations of elementary computational algorithms such as the O(nm) long-form multiplication algorithm chosen for the analysis by Dziri et al.
>
> > It is not clear how much "new insight" this work provides as many of the results have already been reported in other similar contexts.
>
> More generally, our proof tree annotations encode more fine-grained information than previous similar approaches we have seen in the literature. The insights on axiom movements, performance on nonlinear vs. linear problems, and “range prompt” (similar to curriculum learning – see new draft) are new to the best of our knowledge, but we agree this can be emphasized better and we have tried to do so in our new draft.
>
> > Do we know why the LLMs struggle so much in the nonlinear problems (e.g., Figure 3 of the paper) even when depth is small?
>
> That's an interesting question! Nonlinear problems with depth 3 have the same number of leafs (i.e., width) as linear problems with depth 9 and width 10. Comparing the graphs we see a lot lower performance for nonlinear ones with the same depth/width in most cases. This suggests that what makes the problem harder is either that the intermediate conclusions need to be kept longer in memory (a feature of nonlinear problems) or that the comp-eq logical form is more difficult. We looked at some of the reasoning traces and found support for both of these explanations. We did not comment on this in the submitted version but it is a nice addition, so we have added a discussion at the end of 5.3.
>
> > it is not clear why the Llama3-70B model has a near-perfect score on "linear depth" scenarios: is it because of its training data, scale, or something else?
>
> The reason why Llama3 70B performs perfectly is most likely scale, since it does not differ from Llama 8B in training method or data. We have now included a comment on this in the draft.
>
> 2. We actually initially considered measuring the effect of the number of in-context examples in our experiments. Some of our preliminary tests showed that after a small number of examples were shown (< 12), increasing this number did not improve performance. Given the findings in other papers such as Agarwal et al. demonstrating similar findings in a more rigorous setting, we decided not to investigate this experimental variable further. Thank you for raising this point – we have clarified it in the draft, see footnote 7.
>
> 3. [See responses below]
>
> > The paper does not mention how large the overall set of proper names and values were.
>
> Thanks for spotting this! By default our data set of names contains 52 English-language names. For problem types requiring more agents, we use a separate dataset containing 4945 English-language names. The quantities in the axioms of our problems are instantiated in the range [2,20] and constrained such that no intermediate/final quantity of any predicate may exceed 1000 (including the answer). Due to the required arithmetic computations being addition between relatively small numbers, it is unlikely that the degradation in accuracy we observe is due to pure arithmetic errors. We have clarified these points in the draft (see footnote 5) and also added a note on the token bias to the limitations.
>
> > how many of the mistakes were arithmetic mistakes versus logical mistakes.
>
> Parsing the nature of the mistakes is definitely interesting, but nontrivial to do. We therefore leave that for future work. However, by eyeballing it seems that the vast majority of errors are logical rather than numerical – see end of 5.4.
>
> 4. Yes, as you point out, models perform errors even in these simple settings which we can study in a principled way using MathGAP. Moreover, the logical forms follow a commonly accepted taxonomy of arithmetic concepts (Riley et al., 1983) and should thus have a good coverage of standard grade school math problems.

---

> > ### Author Response · Authors · 2024-11-22
> >
> > > I believe this work has potential to have a significant contribution if it can provide more insights and explanations that lead to better understanding of LLMs.
> >
> > We thank you for your feedback which has helped improve the paper and hope that we have addressed some of your concerns in our responses above. On another note, we have added some experiments on the o1 model which are presented in the current draft (see App. C); we find, e.g., that o1 is very sensitive to premise orderings for complex problems. We would also like to emphasize that one of the main contributions of the paper is the evaluation framework itself, which will be open-sourced and we think might lead to even more new insights beyond the present work. We have already started working on some follow-ups, e.g., looking at how generalization interacts with training and whether a model can learn to produce student misconceptions.

---

> ### Comment · Reviewer_tyGU · 2024-11-22
> **Update During the Discussion Period**
>
> I would like to thank the authors for their informative response that clarified several questions and comments. Hence, I raised my score.

---

### Official Review · Reviewer_w9Zb · 2024-11-12

**Soundness:** 3
**Presentation:** 3
**Contribution:** 3
**Rating:** 8
**Confidence:** 3

**Summary:**

This paper introduces MathGAP, a framework designed to evaluate large language models (LLMs) on mathematical word problems requiring proofs of arbitrary complexity. The authors address two critical issues in current evaluations: data contamination due to overlapping training and test sets, and benchmarks not reflecting the arbitrary complexity of problem proofs.

MathGAP uses proof trees characterized by properties such as depth, width, linearity, and ordering, enabling the generation of synthetic problems with controlled complexity and corresponding chain-of-thought (CoT) reasoning annotations. By systematically varying these properties, the framework assesses how well various LLMs generalize to problems more complex than those encountered during training or provided in-context.

The authors conduct comprehensive experiments that reveal model performance significantly declines as proof complexity increases, particularly for nonlinear proofs. Interestingly, providing in-context examples from the same distribution as the test set does not always enhance performance. The study sheds light on the limitations of current LLMs in handling complex reasoning tasks and offers insights into their generalization capabilities.

**Strengths:**

Addresses a Critical Gap: Tackles the underexplored area of evaluating LLMs on problems with arbitrary proof complexity. Directly addresses issues of data contamination and limited complexity in existing benchmarks.

Formalism and Clarity: Utilizes proof trees and logical forms to precisely characterize problem complexity. Well-structured and clearly written, with helpful figures enhancing understanding.

Comprehensive Experiments: Conducts extensive experiments across multiple dimensions of proof complexity (depth, width, linearity, ordering). Evaluates various state-of-the-art LLMs, providing valuable insights into model behaviors and limitations.

Insights into LLM Limitations and In-Context Learning: Reveals that model performance degrades with increasing complexity. Provides meaningful insights into the impact of different in-context learning strategies. Highlights the models' sensitivity to problem structure and ordering.

**Weaknesses:**

Synthetic Data Limitations and Linguistic Diversity: Synthetic problems may not capture the full linguistic and conceptual diversity of real-world problems. Reliance on templates could lead to repetitive linguistic patterns.

Assumption of Reasoning Alignment: Assumes LLMs reason in ways closely aligned with the proposed proof trees. Models might use heuristics or patterns not captured by the formalism, affecting evaluation accuracy.

Limited Error Analysis: Lacks a deep analysis of error types made by the models. More detailed error categorization could provide insights into reasoning limitations (e.g., arithmetic vs. logical errors).

**Questions:**

Have you considered incorporating paraphrasing techniques or using LLMs to generate more diverse linguistic expressions to increase variety and challenge the models? Incorporating more arithmetic relations, irrelevant information, larger numbers, or units to enrich tasks could help increase the diversity as well.

How do you ensure that the proof tree structures accurately reflect LLMs' reasoning processes? Is there a risk that models might not utilize the intended inference paths?

Can you provide more detailed analyses to determine whether performance drops are due to arithmetic computation errors, logical reasoning mistakes, or other factors?

---

> ### Author Response · Authors · 2024-11-22
>
> Thank you for your valuable and positive feedback. We respond to your questions below.
>
> > Have you considered incorporating paraphrasing techniques or using LLMs to generate more diverse linguistic expressions to increase variety and challenge the models? Incorporating more arithmetic relations, irrelevant information, larger numbers, or units to enrich tasks could help increase the diversity as well.
>
> We did consider it, but opted against it as our aim was not necessarily to construct a dataset that is maximally challenging, but to evaluate the effect of particular types of proof complexity on performance. Moreover, LLM paraphrasing might lead to problems that are unfaithful to their specifications. Some of these factors are definitely interesting to explore for future work though.
>
> > How do you ensure that the proof tree structures accurately reflect LLMs' reasoning processes? Is there a risk that models might not utilize the intended inference paths?
>
> There are indeed no guarantees that language models will follow the same reasoning procedure, but this should not affect our evaluation as we only consider answer accuracy. However, by eyeballing some of the outputs, we do observe that they often follow the same post-order traversal as our annotations. We agree that investigating the reasoning traces that LLMs use in a principled manner is an interesting topic for future work, and we note that our framework would allow for such analysis to be carried out.
>
> > Can you provide more detailed analyses to determine whether performance drops are due to arithmetic computation errors, logical reasoning mistakes, or other factors?
>
> Again by eyeballing it seems that the vast majority of errors are logical rather than computational. That is, the model would output an incorrect expression, rather than a correct expression that is incorrectly solved. We have added a comment on this at the end of 5.4 but we did not present any thorough analysis since parsing the nature of the mistakes is nontrivial due to high degree of variability in the outputs. We therefore leave that as an interesting direction for future work.

---

### Author Response · Authors · 2024-11-22

We would like to thank all reviewers for their valuable feedback which has helped improve the paper. We have updated our draft to incorporate many of the points made by the reviewers; please see our responses below and the updated draft for details. Perhaps most notably, we have added experiments on OpenAI o1 as suggested by reviewer iPba.

---

### Meta-Review · Area_Chair_nMtH · 2024-12-21

**Metareview:**

(a) Scientific Claims and Findings:
The paper introduces MathGAP, a framework for evaluating LLMs on arithmetic word problems with arbitrarily complex proofs. Key findings include:
- Most tested LLMs show significant performance degradation as proof complexity increases, particularly with deeper and wider proof structures
- Nonlinear proof structures are especially challenging, even for advanced models like GPT-4o
- Counter-intuitively, providing in-context examples from the same distribution as test data isn't always beneficial for performance
- Zero-shot prompting and demonstrating diverse but less complex examples sometimes perform similarly or better than IID examples
- Models show sensitivity to the ordering of problem premises, suggesting limitations in their reasoning capabilities

(b) Strengths:
- Addresses Critical Evaluation Gaps: The work tackles two major issues in current LLM evaluation - data contamination and limited complexity benchmarking.
- Rigorous Framework: MathGAP provides a systematic way to generate problems with controlled complexity and corresponding chain-of-thought reasoning annotations. The proof trees are well-characterized by properties like depth, width, linearity, and ordering.
- Comprehensive Experimentation: The study includes extensive testing across multiple dimensions of proof complexity and various state-of-the-art LLMs, providing valuable insights into model behaviors and limitations.
- Well-Presented: The paper is clearly written with helpful visualizations and thorough explanation of the methodology.

(c) Weaknesses:
- Limited Diversity: The synthetic nature of the problems may not capture the full linguistic and conceptual diversity of real-world mathematical problems.
- Scope of Analysis: The framework currently uses a relatively constrained set of predicates and inference rules, though these follow established taxonomies.
- Depth of Error Analysis: While some initial observations are made about logical vs arithmetic errors, a more detailed categorization and analysis of error types could provide deeper insights into model limitations.
- Reasoning Alignment Assumptions: The framework assumes LLMs reason in ways aligned with the proposed proof trees, though models might use different heuristics.

(d) Reasons for Accept:
- The paper makes a significant contribution by providing a systematic framework for evaluating LLM reasoning capabilities, addressing important gaps in current evaluation methods.
- The findings provide valuable insights into the limitations of current LLMs in handling complex reasoning tasks.
- The framework is well-designed and enables future research on model behavior and improvement.
- The work has been improved through the review process, including additional experiments with newer models like o1 that strengthen the conclusions.

**Additional Comments On Reviewer Discussion:**

During the review process, reviewers raised concerns about the absence of recent SOTA models, depth of analysis, and framework constraints. The authors addressed these by adding experiments with the o1 model, clarifying distinctions from prior work, and providing additional analysis of error types and performance differences between linear and nonlinear problems. These improvements, along with the paper's significant contribution to systematic evaluation of LLM reasoning capabilities, led to its acceptance.
The work is particularly valuable as it not only provides insights into current LLM limitations but also establishes a framework for future research on model behavior and improvement.

---

### Decision · Program_Chairs · 2025-01-22

Accept (Poster)